# RECOG RL01: Correcting GRACE total water storage estimates for global lakes/reservoirs and earthquakes

Simon Deggim[1], Annette Eicker[1], Lennart Schawohl[1], Helena Gerdener[2], Kerstin Schulze[2], Olga Engels[2], Jürgen Kusche[2], Anita T. Saraswati[3], Tonie van Dam[4], Laura Ellenbeck[5], Denise Dettmering[5], Christian Schwatke[5], Stefan Mayr[6], Igor Klein[6], Laurent Longuevergne[7]

[1]Geodesy & Geoinformatics, HafenCity University Hamburg, D-20457, Germany
[2]Institute of Geodesy and Geoinformatics, University of Bonn, D-53012, Germany
[3]Department of Engineering, University of Luxembourg, L-4364, Luxembourg
[4]Interdisciplinary Centre for Security, Reliability and Trust, University of Luxembourg, L-1359, Luxembourg
[5]Deutsches Geodätisches Forschungsinstitut, Technical University of Munich (DGFI-TUM), D-80333, Germany
[6]Earth Observation Center, German Aerospace Center (DLR), Oberpfaffenhofen, D-82234, Germany
[7]CNRS, Geosciences Rennes - UMR 6118, Université de Rennes, F-35000, France

*Correspondence to*: Simon Deggim (simon.deggim@hcu-hamburg.de)

**Abstract.** Observations of changes in terrestrial water storage obtained from the satellite mission GRACE (Gravity Recovery and Climate Experiment) have frequently been used for water cycle studies and for the improvement of hydrological models by means of calibration and data assimilation. However, due to a low spatial resolution of the gravity field models spatially localized water storage changes, such as those occurring in lakes and reservoirs, cannot properly be represented in the GRACE estimates. As surface storage changes can represent a large part of total water storage, this leads to leakage effects and results in surface water signals becoming erroneously assimilated into other water storage compartments of neighboring model grid cells. As a consequence, a simple mass balance at grid/regional scale is not sufficient to deconvolve the impact of surface water on TWS.

Furthermore, non-hydrology related phenomena contained in the GRACE time series, such as the mass redistribution caused by major earthquakes, hamper the use of GRACE for hydrological studies in affected regions.

In this paper, we present the first release (RL01) of the global correction product RECOG (REgional COrrections for GRACE), which accounts for both the surface water (lakes & reservoirs, RECOG-LR) and earthquake effects (RECOG-EQ). RECOG-LR is computed from forward-modelling surface water volume estimates derived from satellite altimetry and (optical) remote sensing and allows both a removal of these signals from GRACE and a re-location of the mass change to its origin within the outline of the lakes/reservoirs. The earthquake correction RECOG-EQ includes both the co-seismic and post-seismic signals of two major earthquakes with magnitudes above 9 Mw.

We can show that applying the correction dataset (1) reduces the GRACE signal variability by up to 75% around major lakes and explains a large part of GRACE seasonal variations and trends, (2) avoids the introduction of spurious trends caused by leakage signals of nearby lakes when calibrating/assimilating hydrological models with GRACE, even in neighboring river basins, and (3) enables a clearer detection of hydrological droughts in areas affected by earthquakes. A first validation of the

corrected GRACE time series using GPS-derived vertical station displacements shows a consistent improvement of the fit
between GRACE and GNSS after applying the correction. Data are made available as open access via the Pangea database
(RECOG-LR: Deggim et al. (2020a), https://doi.org/10.1594/PANGAEA.921851; RECOG-EQ: Gerdener et al. (2020b, under
revision), https://doi.pangaea.de/10.1594/PANGAEA.921923).

## 1 Introduction

The dynamic global water cycle influences our everyday lives by affecting freshwater availability, weather/climate fluctuations
and trends, seasonal variations, anthropogenic water use, and single extreme events such as floods and droughts. Understanding
how water is transiently stored and exchanged among the different compartments (groundwater, surface water, soil moisture,
etc.) with the help of hydrological models is, therefore, of major societal importance. However, large model uncertainties
caused by errors in climate forcings and an incomplete realism of process representations limit the models' ability to accurately
simulate water storages and fluxes making independent observations for model validation/calibration and data assimilation
indispensable.

Since 2002 measurements of time variable gravity obtained with the twin-satellite mission GRACE (Tapley et al., 2004) and
its successor mission GRACE-Follow-On (GRACE-FO, Flechtner et al. 2016) have allowed for the determination of column-
integrated terrestrial water storage (TWS) changes on the global scale with uniform data coverage (e.g. Pail et al., 2015).
However, several challenges are involved with using GRACE for improving hydrological models, among them (1) the low
spatial resolution of GRACE, integrating spatially over regions as large as ~200 000 km² and hampering the representation of
concentrated and sub-scale water storage changes and (2) the fact that gravity observations contain also non-hydrology related
mass variations.

The first problem is due to the GRACE orbit configuration in combination with unmodelled short-periodic mass changes,
resulting in the gravity field models being strongly corrupted by spatially correlated noise. The necessary spatial filtering
approach (e.g. Kusche 2007) inevitably leads to signal loss and to leakage effects resulting in a rather coarse spatial resolution
of the gravity field models of a few 100 km.

This limits the investigation of mass variations to rather large-scale processes (Longuevergne et al., 2013), even though small-
scale mass variations, such as caused by human-controlled reservoirs or strong seasonal variations of natural lakes, whose
typical size is smaller than GRACE resolution but large enough in magnitude, can have a strong influence on the total mass
change signal (Frappart et al., 2012). Even though GRACE can "see" these mass changes, they do not necessarily appear
exactly at the location of their origin and with the correct magnitude. Thus they can distort the water storage estimate for
neighboring areas or the average over a river basin, as shown in Fig. 1.

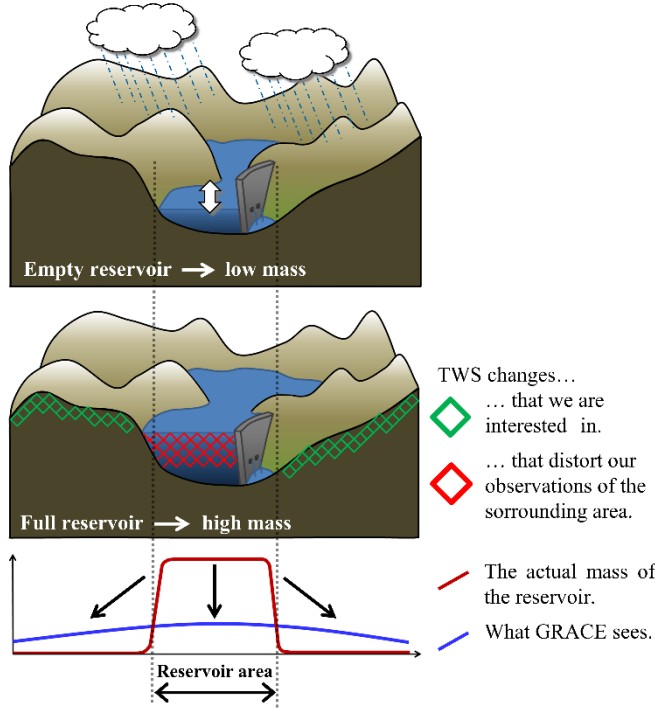

**Figure 1: Overview of the lake leakage problem with localized changes in water level of the lake/reservoir influencing the estimated water storage in surrounding areas.**

This subscale mass variability impacts GRACE amplitudes up to 20% averaged over basins as large as ~200,000 km² (Longuevergne et al. 2013, Farinotti et al. 2015). Although the issue of concentrated mass is general in hydrology and not limited to surface water bodies (e.g. Castellazzi et al., 2018), dam operations and impoundment have a large impact on the water cycle and on continent-ocean exchanges (Chao et al., 2008). Therefore, several publications where specifically interested in removing the impact of surface water bodies on GRACE total water storage changes for further studies. For example, Grippa et al. (2011) removed the influence of surface water storage in the Niger River (derived from altimetry and remotely sensed surface water extent) from GRACE TWS estimates to better be able to compare them to hydrological model output. Tseng et al. (2016) determined mass changes in two Tibetan lakes combining altimetry, remote sensing and GRACE estimates and Zhang et al. (2017) estimated water volume changes for 96% of the lake area on the Tibetan plateau by combining an average of ICESat-derived elevation changes from a number of larger lakes with Landsat lake area changes for smaller lakes. Ni et al. (2017) removed leakage error in GRACE estimates over Lake Volta basin in Ghana using constrained forward-modelling. All these studies represent regional test cases but a global assessment of the influence of surface water body mass change on GRACE data is missing.

However, using GRACE data for the evaluation of (global) hydrological models or for combining models and observations by model calibration (Werth & Güntner 2010) and data assimilation (C/DA, Zaitchik et al., 2008, Eicker et al., 2014) without accounting for localized surface water storage can lead to two different kind of errors. (i) Many global hydrological models do





not include a surface water storage compartment at all (Scanlon et al., 2017) and assimilating GRACE TWS into a model that does not explicitly include surface waters will inevitably result in other storage compartments (such as soil moisture or groundwater) to become distorted by absorbing the observed surface water mass change. Even if a model does include a surface

water compartment (Müller Schmied et al., 2014a), it might not represent the realistic behavior of, e.g. man-operated reservoirs and it might be preferable to exclude the reservoir storage from the assimilation. (ii) The leakage effect of localized surface water bodies might cause an assimilation of the surface water mass change into neighboring grid cells that should not be affected by it. To our knowledge, no investigation so far has studied the effects of surface water bodies on GRACE model calibration or data assimilation and how they can best be handled in order to not distort the C/DA results. Having a global

correction dataset to clear GRACE water mass changes of the influence of large surface water bodies (here: lakes and reservoirs) will be immensely helpful for making GRACE estimates more consistent with model output.

Today, extensive information on surface water variations is available from satellite remote sensing. For almost 30 years, satellite altimetry has been providing water levels of large and medium lakes and reservoirs on a global scale (e.g. Birkett, 1995; Berry et al., 2005; Göttl el al., 2016). Several databases make these time series freely available for hydrological

applications, among them the Database for Hydrological Time Series of Inland Waters (DAHITI; Schwatke et al., 2015). In addition, optical satellite images are used to derive surface extent of lakes and reservoirs (e.g. Pekel et al., 2016; Klein et al.,2017; Schwatke et al., 2019; Schwatke et al., 2020). Time series from optical sensors can reach a length of up to almost 40 years with spatial resolution between 250 m (MODIS), 30 m (Landsat) and 10 m (Sentinel-2) as well as high temporal resolution with revisit time from 14 days (Landsat), over 5 days (Sentinel-2), and up to 1 day (MODIS). By combining height

and surface area information, time series of storage changes can be derived purely based on remote sensing data (e.g. Busker et al., 2019, Crétaux et al., 2011).

To account for the second challenge in using GRACE data for hydrological studies, namely the removal of all non-hydrology-related mass variations, some effects are typically subtracted using geophysical models either during the computation of the gravity field solutions (e.g. Earth tides, ocean tides, and oceanic/atmospheric mass variations) or in post-processing (e.g. glacial

isostatic adjustment (GIA)). However, in addition to this, also the mass redistribution caused by the crustal deformation following large earthquakes is contained in the GRACE observations masking hydrological phenomena in the affected regions. Several studies highlighted GRACE's usefulness for estimating large earthquakes (magnitude above 9.0), e.g. Panet et al. (2007), Broerse (2014), Einarsson et al. (2010), Einarsson (2011), and Wang et al. (2012) and have also identified co- and post-seismic earthquake signals in GRACE data with a lower magnitude, e.g. Han et al. (2016) and Zhang et al. (2016) down

to magnitude 8.3 (Chao and Liau, 2019). For example, the onset of the Sumatra-Andaman earthquake in December 2004 (magnitude 9.1) was analyzed using, among others, differences of monthly gravity solutions (Han et al. 2006), wavelet analysis (Panet et al., 2007), Bayesian approaches (Einarsson et al., 2010), or normal modes (Cambiotti et al., 2011). At time of writing, the German GeoForschungsZentrum in Potsdam (GFZ) is the only processing centre that provides a total water storage (level 3) dataset corrected for earthquakes (Boergens et al., 2019), however, a data-based global earthquake correction for different

GRACE solutions is not available yet. To account for both the localized surface water storage in lakes/reservoirs and the



earthquake signal, we present a new global correction dataset RECOG RL01 which can be used for disaggregation of the integral GRACE water storage estimates in addition to applying standard corrections such as GIA models and the atmosphere/ocean de-aliasing products. The term RECOG refers to the fact, that all effects included in the data product are localized phenomena that nevertheless influence a larger region around them. The surface water correction (RECOG-LR) was
computed from forward-modelling altimetry and remote sensing observations and can be used to (a) subtract the lake/reservoir storage from the GRACE time series (removal approach) and (b) to relocate the surface water storage at its exact location of origin (relocation approach). The earthquake correction (RECOG-EQ) was estimated from GRACE monthly solutions using the Bayesian approach provided in Einarsson et al. (2010) and takes into account both the co-seismic and the post-seismic signal.

This paper is organized as follows: Section 2 describes the input data, pre-processing steps and the forward-modelling procedure. Section 3 presents the resulting correction products and visualizes some of its key characteristics. In Section 4 the influence of the dataset is shown for exemplary applications: data assimilation of GRACE into the global hydrological model WaterGAP, the detection of drought indices in an earthquake affected region and a validation of the correction product using GNSS-observed station displacements. This is followed by a discussion of the benefits and limitations of the correction
product. Section 5 summarizes the findings and gives an outlook to further development options.

## 2 Methods

In this section we describe the various input data and their sources (Sec. 2.1.1 and 2.1.2) that were used to perform the forward-modelling (Sec. 2.1.4) of the surface water bodies and its necessary pre-processing steps (Sec. 2.1.3). The earthquake correction will be handled in Section 2.2.

### 2.1 Lake/reservoir correction RECOG-LR

The lake/reservoir correction is based on a subset of currently 283 of the largest surface water bodies monitored with satellite altimetry. The lake water volume variations product is designed around (1) monthly water level time series from a global multi-satellite product and (2) surface water extend area for each lake.

### 2.1.1 Lake level time series from altimetry

Satellite altimetry measures the distance between the satellite and the Earth surface (i.e. the range) by analyzing the transmitted and received radar echo after it has been reflected by the Earth's surface. Originally, the technique was developed for open ocean applications. However, if the data is carefully pre-processed, it can also be used for estimating the height of inland water bodies such as lakes and reservoirs. Since the inland signals are frequently contaminated by land reflections, a rigorous outlier detection (Schwatke et al., 2015) as well as a dedicated retracking (e.g. Passaro et al., 2018) is mandatory.





In this study, time series created by DAHITI (Schwatke et al., 2015) are used. DAHITI provides water level time series of more than 2000 globally distributed inland targets, i.e. lakes, rivers, and reservoirs in a period between 1992 and today, depending on the satellite mission covering the water body. The temporal resolution of the time series differs depending on the size of a lake. Small lakes that are only covered by one single satellite track can be measured every 35 or 10 days (depending on the mission), whereas for large lakes a height can be derived almost every day. Moreover, information can only be provided

for those water bodies located directly beneath a satellite's tracks (Dettmering et al., in review), preventing the creation of water level time series of small lakes located between the satellites ground tracks. In addition, for small lakes or lakes surrounded by large topography no reliable height information might be created due to corrupted or too noisy radar echoes. The quality of the DAHITI water level time series depends on various criteria, mainly on the size of the lake and the length of the crossing satellite track as well as the surrounding topography. Comparison with in-situ data show RMSE of a few

centimeters for larger lakes and RMSE of some decimeters for river crossings (Schwatke et al., 2015).

### 2.1.2 Creating lake shapes from Remote Sensing

Based on Moderate-resolution Imaging Spectroradiometer (MODIS) optical satellite data, daily surface water extents are provided by the DLR's Global WaterPack (GWP) product (Klein et al., 2017). To receive reliable estimates of the extent of large global lakes and reservoirs, daily observations are aggregated to obtain maximum waterbody extents for the years 2003

to 2018. To capture coherent waterbodies, a pixel based region-growing algorithm is applied, using ancillary information of the temporal static HydroLAKES dataset (Messager et al., 2006) for waterbody identification. Hereby, every water pixel in the aggregated GWP raster layer that spatially overlaps with the original HydroLAKES shape file is assigned to the lake ID given by the HydroLAKES database. Subsequently, a seed point in every designated waterbody is determined, from which an 8-pixel search window region growing is initiated, thus identifying neighboring water pixels. This ensures that waterbodies

are represented by coherent pixel groups only. After the growing process is finished, results are vectorized. With this dynamic approach, the risk of over- or underestimation of the actual water surface extent is reduced (see Fig. S1 in Supplement for further details).

### 2.1.3 Data pre-processing

The input data were combined taking into account several pre-processing steps: months with several data points in the water

level time series were averaged to a monthly mean for consistency with the temporal gravity field resolution. Missing months were linearly interpolated. The water level time series were cut to the investigation period (January 2003 – December 2016) and then reduced by their respective means. To ensure a quick update of the correction product when new lakes will be added to the source databases or when their time series will be updated, the algorithm to match water level time series with their respective lake surface area (as well as most of the following workflow) was automated. The first step for the combination was

an automatically generated data table with global common lake IDs. Where no IDs were available, matching was achieved by comparing names while making sure that no double naming occurred in the input data. If that was not possible or no names





were given, matches were found using a very strict search algorithm for the nearest lake shape to a given time series. If none of the above methods was successful, the respective lake was dismissed and not included in the correction. In total, matches for 283 lakes were found for RECOG-LR RL01 (see Fig. 2), a detailed list is provided in the Supplement (Sec. S1). The surface

water body shapes were then discretized on a fine resolution 0.025° grid to be able to capture long but narrow reservoirs in valleys. Multiplication with the altimetry-derived water height resulted in water volume estimates for each of the grid cells, that were subsequently distributed proportionally over a lower resolution 0.5° grid. The resulting global grids of lake/reservoir-related water height anomalies for each of the 168 months of our investigation period then entered the forward-modelling algorithm.

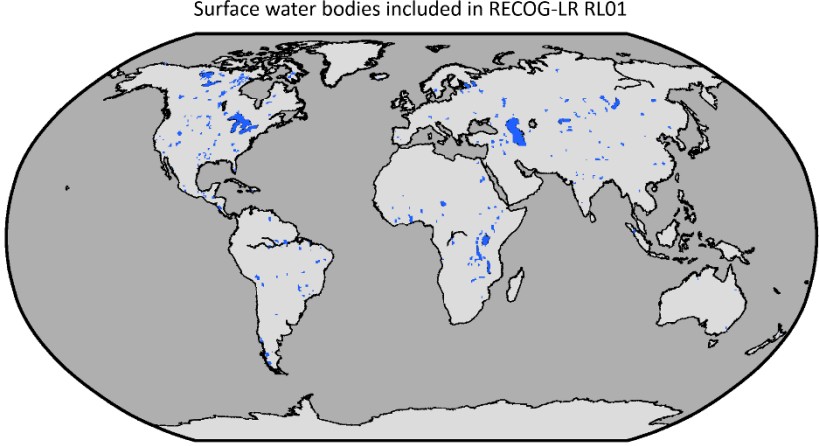


**Figure 2: Overview of all 283 lakes/reservoirs in the dataset (blue areas) given on a 0.5° grid.**

### 2.1.4 Forward-Modelling

The localized altimetry/remote sensing-derived surface water variations have to be converted to the GRACE spatial resolution before they can be subtracted from monthly GRACE gravity field estimates. In this forward-modelling step, the gridded values

were expanded into spherical harmonic coefficients up to degree ($n$) and order ($m$) 96 according to,

$$\begin{bmatrix} \Delta C_{nm} \\ \Delta S_{nm} \end{bmatrix} = \frac{R^2}{M} \cdot \frac{k_n + 1}{2n + 1} \int_0^\pi \int_0^{2\pi} \Delta TWS(\theta, \lambda) \cdot P_{nm}(\cos \theta) \cdot \begin{bmatrix} \cos(m\lambda) \\ \sin(m\lambda) \end{bmatrix} \cdot \sin(\theta) \cdot d\lambda \cdot d\theta \qquad (1)$$

with $\Delta C_{nm}$ and $\Delta S_{nm}$ being the spherical harmonics coefficients (at degree $n$ and order $m$), $R$ the radius of the earth, $M$ the mass of the earth, $k_n$ the load Love numbers (Farrell, 1972), $\Delta TWS(\theta, \lambda)$ the changes in altimetry-derived total water storage in dependence on colatitude $\theta$ and longitude $\lambda$, and $P_{nm}$ the Legendre functions. Equation (1) was discretized on a global 0.025° x 0.025° grid and we subsequently apply a standard GRACE filter for smoothing (DDK3, Kusche, 2007; Kusche et al.,

2009). This gives us the idealized signal that GRACE would measure if it was influenced by the changing mass in the lakes/reservoirs only. For a grid-based evaluation (0.5° x 0.5° grid) a recomputation using Eq. (2) is necessary to calculate the lake water storage for every grid cell after filtering (Wahr et al., 1998):



$$\Delta TWS(\theta, \lambda) = \frac{M}{4\pi R^2 \rho} \sum_{n=0}^{96} \sum_{m=0}^{n} \frac{2n+1}{1+k_n} \cdot P_{nm}(\cos\theta) \cdot (\Delta C_{nm} \cos(m\lambda) + \Delta S_{nm} \sin(m\lambda)) \tag{2}$$

Here we included the density of water $\rho$ ($1025 \frac{\text{kg}}{\text{m}^3}$) to get a total water storage result in meters of equivalent water heights, corresponding to the input variations in total water storage from the water level time series.

To allow for various applications, we provide two different results that are explained in Fig. 3, namely (1) the forward-modelled lake water correction to be subtracted from the GRACE data to remove the influence of lakes/reservoirs (removal approach). It is provided both on spherical harmonic level (Eq. 1) and as a gridded data product (Eq. 2) in terms of TWS. The relocation approach (2) aims for conserving the lake water mass and restore its original water surface area extent by subsequently re-adding the altimetry-derived water height changes for each lake respectively.

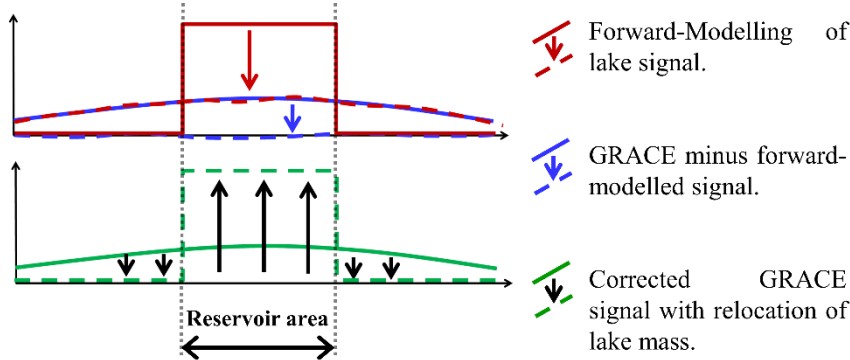


Figure 3: Scheme of forward-modelling approach. Removal (top) and relocation (bottom) of lakes and reservoirs.

## 2.2 Earthquake correction

Different from the lake/reservoir correction, which is computed by forward-modelling using independent datasets (altimetry/remote sensing), the Earthquake correction is derived by fitting a parametric function to monthly GRACE data along 210 the following processing line: In a first step, spherical harmonic coefficients have to be backward-modelled to gridded geoid changes by

$$\Delta GC(\theta, \lambda) = R \sum_{n=0}^{96} \sum_{m=0}^{n} P_{nm}(\cos\theta) \cdot (\Delta C_{nm} \cos(m\lambda) + \Delta S_{nm} \sin(m\lambda)) \tag{3}$$

to be able to apply the Bayesian approach provided in Einarsson et al. (2010). The total geoid changes for a specific location $(\theta_i, \lambda_j)$ can be subdivided into a bias ($\Delta GC_{bias}$), trend ($\Delta GC_{trend}$), annual- ($\Delta GC_{ann}$), and semi-annual signal ($\Delta GC_{semiann}$), S2 aliasing period of 161 days ($\Delta GC_{N2}$) and the earthquake signal ($\Delta GC_{EQ}$) as

$$\begin{aligned} \Delta GC(\theta_i, \lambda_j, t) = &\Delta GC_{bias}(\theta_i, \lambda_j, t) + \Delta GC_{trend}(\theta_i, \lambda_j, t) + \Delta GC_{ann}(\theta_i, \lambda_j, t) + \Delta GC_{semiann}(\theta_i, \lambda_j, t) \\ &+ \Delta GC_{S2}(\theta_i, \lambda_j, t) + \Delta GC_{EQ}(\theta_i, \lambda_j, t) \end{aligned} \tag{4}$$





which contains the model coefficients $C_{bias}, C_{trend}, C_{ann}, \phi_{ann}, C_{semiann}, \phi_{semiann}, C_{S2}$ and $\phi_{S2}$. The earthquake signal included here is described by a co-seismic and a post-seismic component modelled as

$$\Delta GC_{EQ}(\theta_i, \lambda_j, t) = C_{v_{co}} H_{t_v}(t) + C_{v_{post}} H_{t_v}(t)\left(1 - e^{-\frac{t-t_v}{\tau}}\right). \tag{5}$$

$C_{v_{co}}$ and $C_{v_{post}}$ describe coefficients for the co- and post-seismic component of the respective earthquake $v$, $H_{t_v}(t)$ is the Heaviside step-function at time $t_v$, and $\tau$ is the decay rate. All coefficients are then estimated using Monte-Carlo integration for quasi-linear models and are used to estimate the total earthquake signal. This signal is then removed from the total geoid changes for each considered earthquake to derive an earthquake-corrected dataset. Furthermore, we applied a spatial radial Gaussian window to consider only regions that were affected by earthquakes. The center of the Gaussian window is placed in the epicenter of the respective earthquake. Einarsson's approach is, as recommended, consecutively applied to earthquakes with a magnitude that is larger or equal 9.0, which in fact are the Sumatra-Andaman earthquake (M9.1) in December 2004 and the Tohoku earthquake (M9) in March 2011. Broerse (2011), for example, showed that earthquakes with a magnitude larger than 9.0 are clearly visible in GRACE data, while earthquakes with lower magnitude cannot always be clearly separated. For more information about the approach see Einarsson et al. (2010) and Einarsson (2011). To derive TWS anomalies, the geoid changes are forward-modelled to spherical harmonic coefficients similar to Eq. (1) by

$$\begin{bmatrix} \Delta C_{nm} \\ \Delta S_{nm} \end{bmatrix} = \frac{1}{R} \int_0^\pi \int_0^{2\pi} \Delta TWS(\theta, \lambda) \cdot P_{nm}(\cos\theta) \cdot \begin{bmatrix} \cos(m\lambda) \\ \sin(m\lambda) \end{bmatrix} \cdot \sin(\theta) \cdot d\lambda \cdot d\theta \tag{6}$$

and again backward-modelled to TWS changes as described in Eq. 2. The final earthquake correction product RECOQ-EQ is then derived by computing the difference between the uncorrected and earthquake-corrected TWSA.

## 3 Correction Datasets

### 3.1 Lake and reservoir correction: RECOG-LR

The resulting product for the lake and reservoir correction consists of three datasets: (1) The monthly correction for each grid cell to remove the lake signal (removal approach), (2) the same given as spherical harmonic coefficients and (3) the altimetry-derived monthly water levels for each grid cell that can be used to re-add the measured lake volume to its actual area (relocation approach). The gridded datasets (1 and 3) have a spatial resolution of 0.5° x 0.5 degrees with global coverage, incorporating data from 283 of the major lakes and reservoirs. The spherical harmonics correction is developed up to degree and order 96 and is smoothed with a standard DDK3 filter to match a typical resolution of monthly GRACE gravity field models (Sec. 2). Differently filtered corrections are available upon request. We cover the time span from January 2003 to December 2016 with a monthly temporal resolution resulting in 168 months of data. Figure 2 highlights each grid cell that includes surface water bodies with data used for the correction. Note that although most of the major lakes and reservoirs are covered, some had to be excluded from the dataset for reasons explained further in Sec. 4.5. Figures showing an exemplary seasonal cycle of





RECOG-LR can be found in the Supplement (Fig. S2) and an animation of the monthly changes of the lake/reservoir water storage for the full time series is provided as Video Supplement (Deggim et al. (2020b), https://doi.org/10.5446/48188).

Figure 4a shows the mean amplitudes of the seasonal variations of the correction for each grid cell. The most prominent

features are the Caspian Sea in Asia and the Great Lakes in North America with mean TWS corrections of about 10 cm. However, peaks for individual months can reach as much as 30 cm of TWS correction. Most of the other lakes and reservoirs have mean correction amplitudes in the area of 0 to 3 cm. Fig. 4b displays the linear trend of the lake correction product. Again, the most distinctive features are a strong negative trend of the Caspian Sea and a strong water storage increase in the Great Lakes (altimetry time series shown in Fig. S3 in the Supplement for comparison). Strongly visible is also a positive trend (mass

increase) in Lake Victoria accompanied by a clear mass loss in the nearby Lake Malawi. However, it also becomes evident that some surface water bodies can have a rather prominent trend signal without having a strong seasonality, as for example Lake Oahe in South Dakota, or exhibit a strong seasonality without any long-term trend, such as Lake Chad in Africa or Lake Guri and Tucurui Reservoir in South America. In other surface water bodies, such as the artificial reservoir Lake Volta, the time series is dominated by a strong inter-annual signal (Ni et al. 2017), which does not show up prominently in Fig. 4, but is

very much visible in the total temporal variability shown in Fig. 6 below.

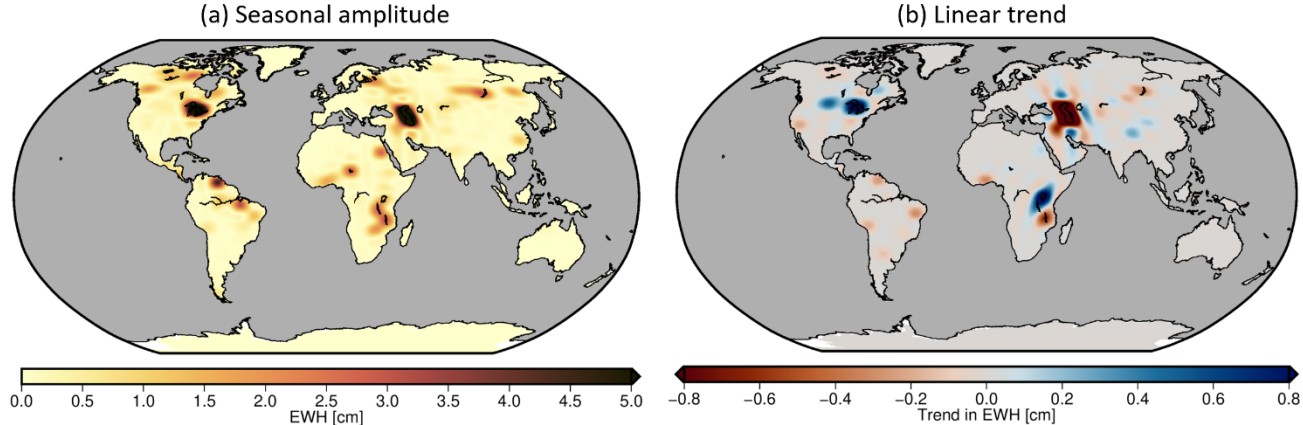

**Figure 4: RECOG-LR on a global scale with its (a) mean seasonal amplitude and (b) trend.**

**3.2 Earthquake correction RECOG-EQ**

The necessary steps for removing the earthquake signal and processing the earthquake correction (RECOG-EQ) from the

GRACE data are applied as described in Sec 2.2. The correction is provided with similar processing steps as the lake correction: The dataset is processed on a global 0.5° grid using spherical harmonic coefficients up to degree and order 96, is DDK3 filtered (different filters are available upon request) and covers the period 2003 to 2016.

As expected, the correction shows only differences over the regions of the 2004 Sumatra-Andaman earthquake (Fig. 5a) and the 2011 Tohoku earthquake (Fig. 5b) because we only corrected for these two earthquakes. For the Sumatra-Andaman region

the linear trends of the correction reach from about -2.7 to 1.1 cm EWH per year. Negative linear trends of down to -2.7 cm EWH per year can be found north of Indonesia and west of the Malaysian peninsular, while positive trends are contained in the Indian Ocean close the coast of Sumatra. Considering Tohoku, the linear trends range from about -2.5 to 1.4. The dominant negative part can be found in the north and eastern of the region Tohoku, while the positive parts are apparent in the Pacific Ocean, southeast of Tohoku. These results let assume that uncorrected TWS changes might hinder correct analysis of the data

for hydrological studies, because the post-seismic part of the earthquake might falsely be interpreted as linear trend in the uncorrected TWS changes, especially when the earthquake is at the beginning of the time series as it is the case for the Sumatra-Andaman earthquake. The results shown here are derived from the ITSG-Grace2018 solutions (Kvas et al. 2019). Earthquake corrections derived from other GRACE solutions provide similar findings, for completeness they are attached in the Supplement (Fig. S4 and S5).

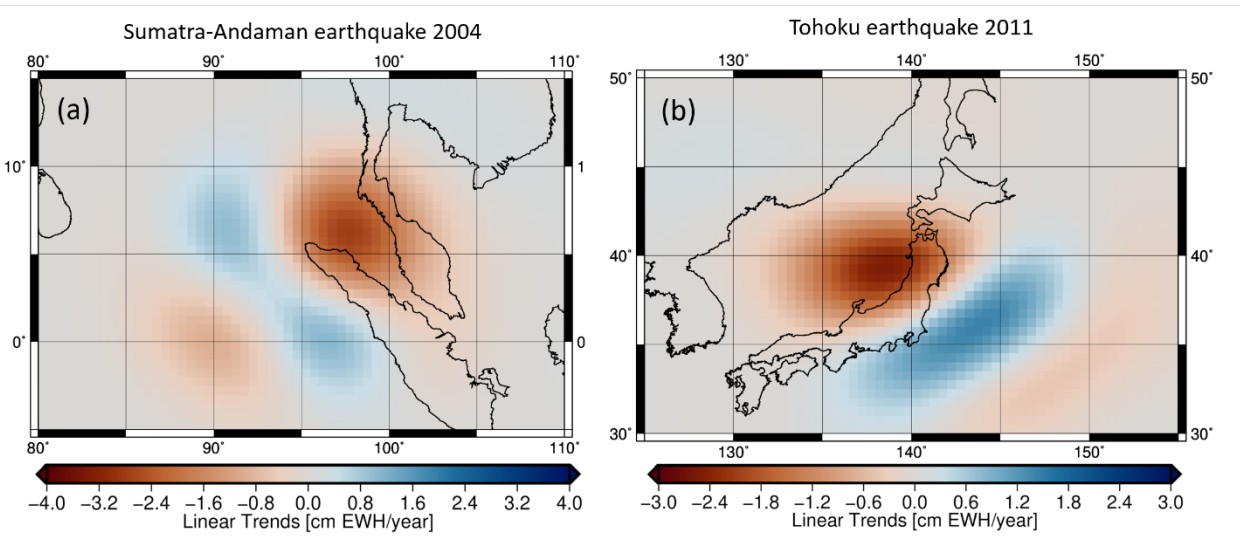


**Figure 5: Linear trends (01/2003 – 12/2016) of TWS changes (cm EWH/year), of the earthquake correction that includes the (a) Sumatra-Andaman earthquake from 2004 and the (b) Tohoku earthquake from 2011.**

## 4 Applications and Discussion

### 4.1 Influence of RECOG RL01 on a global GRACE time series

**4.1.1 Influence of lake/reservoir correction RECOG-LR on GRACE**

We first investigate the influence of subtracting the lake/reservoir correction dataset from a global GRACE time series. For this purpose, we derive gridded TWS anomalies (TWSA) from the ITSG-Grace2018 (Mayer-Gürr et al., 2018, Kvas et al., 2019) spherical harmonics up to degree and order 96 considering corrections for low degree coefficients and glacial isostatic adjustment (Swenson et al., 2008; Cheng et al., 2011, Sun et al., 2016) and applying the DDK3 filter (Kusche, 2007).

For the removal approach, we then reduce the GRACE-derived TWSA grids (Sec. 2) by the lake correction (Sec. 3.1). For the relocation approach, we re-add the altimetry-derived monthly water levels. This leads to three global TWSA datasets: (1)



GRACE-based only, (2) GRACE-TWSA after removing altimetry-based lake/reservoir storage and (3) GRACE-TWS with relocated altimetry-based lake signal.

Figure 6a shows the temporal root means square (RMS) of the lake correction for each grid cell. Subtracting this correction

reduces the temporal RMS variability in the GRACE time series (Fig. 6b) by up to 75% around Caspian Sea in Asia and 50% around Lake Victoria in Africa, compared to the original variability in ITSG-Grace2018. Values around the other lakes vary between 0 and 30% with a few negative values in the area south of the Caspian Sea and in Canada. The later can most likely be attributed to Gibbs oscillations by the bandlimited spectral representation of the data, but also the option that the lake signal was hiding another impact, that can only be recovered after subtracting the correction, should not be completely ruled out, e.g.

in the case of water transfer between compartments as from glacier/snow to lake water (Castellazzi et al., 2019).

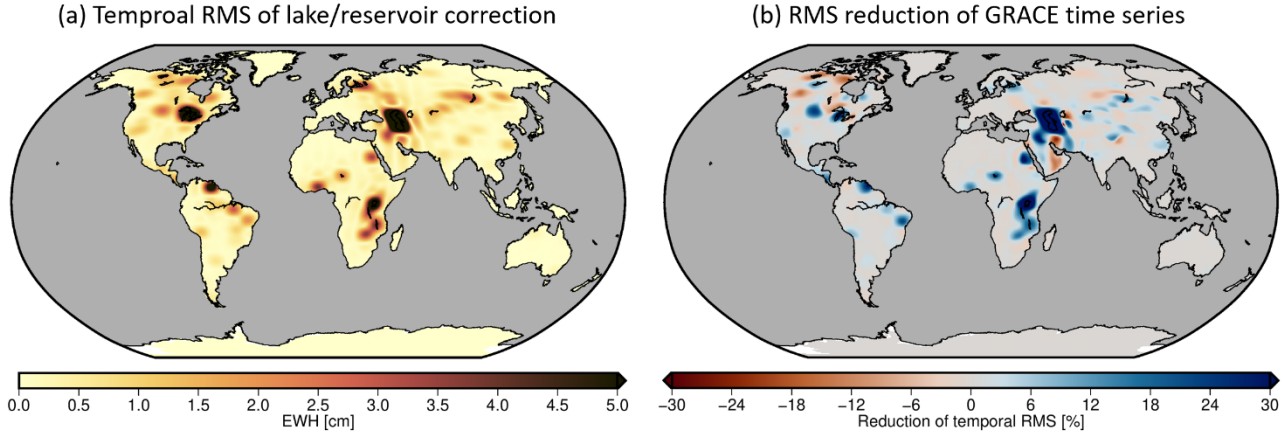

**Figure 6: Temporal root mean square (RMS) of the lake/reservoir correction time series for each grid cell (a) and the relative reduction in temporal RMS when subtracting the correction (removal) from the original GRACE TWS time series (b).**

Figure 7 shows the influence of the lake correction on the linear trend in the GRACE time series for two detailed examples. In

the area around the Caspian Sea (first row of Fig. 7), a very strong negative trend of around -3 cm/year in the original GRACE time series (left) is almost completely removed by the lake correction (middle). The relocation approach then restores the altimetry-derived lake water variation to the lake area (right). The second example shows the Mississippi basin (bottom row of Fig. 7). Even though the Great Lakes are not part of the basin, they still have an effect, particularly on subbasin Alton (everything upstream from Alton, Illinois, USA) that is closest to the Great Lakes and influence the GRACE signal for this

subbasin to up to 5 cm in TWS for some months. This example also shows that a positive trend visible in the original GRACE data can mainly be attributed to surface water change and can be levelled by the correction. Subtracting the lake correction also reduces the positive trend partly caused by smaller lakes/reservoirs (e.g. Lake Oahe and Lake Sakakawea) in the Hermann subbasin in the northwest of the Mississippi.

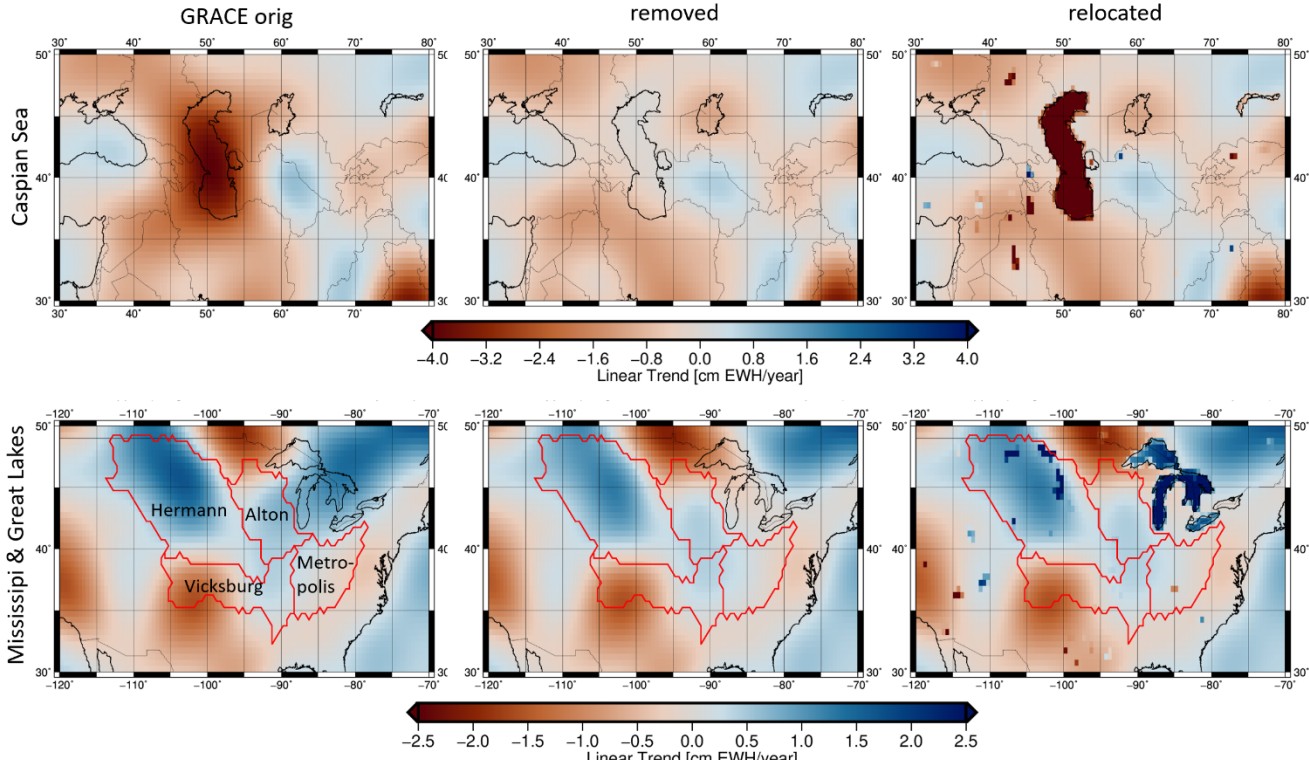

**Figure 7: Linear trend of GRACE-TWS anomalies (01/2003 – 12/2016) in the Caspian Sea region (top) and in the Mississippi basin and around the Great Lakes (bottom) without any correction (left), after removing the influence of lakes (middle) and after relocating the lake signal (right). Please note that the Aral Sea has been excluded from RECOG-LR RL01 due to its strongly varying surface area, which is not yet captured in the database.**

### 4.1.2 Influence of earthquake correction RECOG-EQ on GRACE

This section presents the application of the earthquake correction (Sec. 3.2) on the GRACE data. Linear trends for the period 2003 to 2016 are derived from (1) the original GRACE TWSA and (2) the corrected TWSA after applying RECOG-EQ. Applying the correction changes the spatial pattern of the linear trend in the Sumatra-Andaman region (Fig. 8a and 8c) and the magnitude of positive trends increases by about 0.5 cm EWH per year from 4.2 to 4.7 cm EWH per year. The spatial extension of the positive trends in the corrected data (Fig. 8c) reaches to the Malaysian peninsular. Thus, the former slightly negative trends of about -1 cm EWH per year identify a smaller magnitude or even slightly positive in the earthquake corrected data set. The change in trends might bias the correct analysis of linear trends for this region.

When analysing the Tohoku earthquake results (Fig. 8b and 8d), we also see that magnitude and spatial pattern of the trends change. In this case, the difference between original and corrected GRACE data is more obvious than in the Sumatra-Andaman region: The original dataset shows positive linear trends of about 2 cm EWH per year in the west of Tohoku, while negative trends of about -2 cm EWH per year can be found in the Pacific Ocean along the southeastern coast. These trend signals vanish

almost completely after subtracting the earthquake correction confirming our assumption made in Sec. 3.2: The post-seismic earthquake component was identified as linear trends in the original GRACE data and has clearly biased the trend analysis leading to misinterpretation of the trends, especially for the Tohoku region.

**Figure 8: Linear trends (01/2003 – 12/2016) of TWS changes (cm EWH/year), before and after removing earthquake signals of the Sumatra-Andaman earthquake from 2004 ((a) before, (c) after) and the Tohoku earthquake from 2011 ((b) before, (d) after).**

To analyse the effect of earthquakes on TWS changes in more detail and over land, spatially averaged TWS anomalies are compared for the Malaysian peninsular in Fig. 9a and for Japan in Fig. 9b. Additional figures showing the signal in the epicentre of the two Earthquakes are shown in the Supplement (Fig. S6). Regarding the Malaysian peninsular, the uncorrected TWSA (black) shows a strong decrease in TWS changes beginning in 2004, which results from the Sumatra-Andaman earthquake. After applying the correction (blue), this strong decrease has been removed. The correction (red) shows nicely the co-seismic component of about -2.5 cm EWH as jump between December 2004 and January 2005 and a following post-seismic relaxation, which increases the total correction towards -6 cm. Similar findings can be observed for the results for Japan and the Tohoku

earthquake: The correction contains a co-seismic component down to -6 cm with afterwards post-seismic relaxation but in this case the relaxation is slowly decreasing the amount of correction. However, both examples clarify that an uncorrected GRACE data could lead to wrong conclusions about the underlying signals as the apparent trends can, e.g., hamper the identification of drier and wetter years in the GRACE time series.

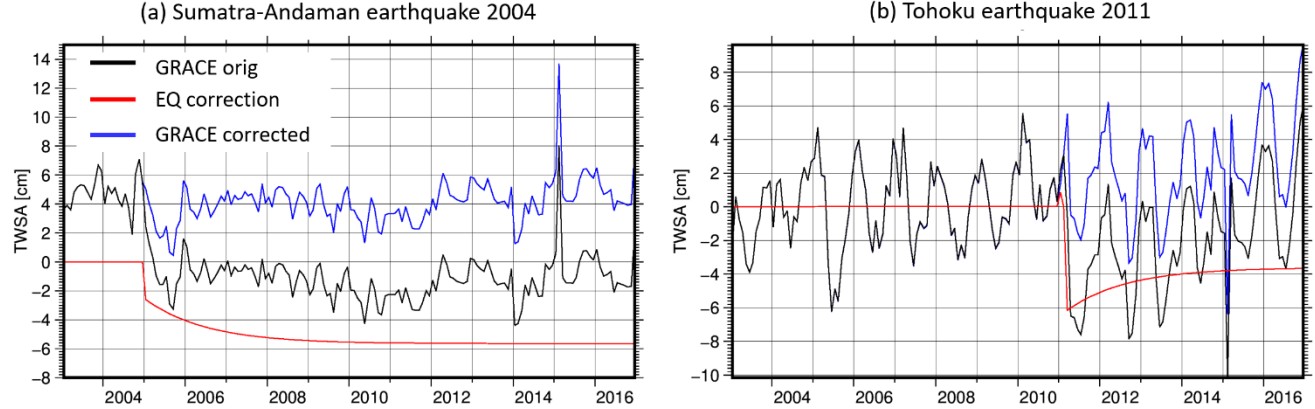

**Figure 9:** **Spatially averaged GRACE TWS anomalies for the original (black) and earthquake corrected (blue) GRACE data and its correction (red) for (a) the Malaysian peninsular (West Malaysia) and (b) Japan.**

## 4.2 Data Assimilation (DA) into hydrological models

An important application of the correction datasets is the removal of lake/reservoir water storage (and earthquake signals) before assimilating GRACE data into hydrological models, if these models either do not include an explicit surface water

component or if strong leakage effects in the GRACE data can be suspected to prevent a reasonable data assimilation.

For a test case in the Mississippi river basin we assimilate subbasin-averaged TWSA time series of all three GRACE datasets (original, removed and relocated) into the Water-Global Assessment and Prognosis (WaterGAP; Döll et al., 1999; Alcamo et al., 2003; Müller Schmied et al., 2014b) hydrological model. WaterGAP is a global hydrological model simulating ten water compartments (snow, canopy, soil, groundwater, global/local lakes, global/local wetlands, reservoirs, and river) for each grid

cell of a global 0.5° x 0.5° grid considering human influences on water resources. Our assimilation theory follows Eicker et al. (2014) and Schumacher et al. (2015, 2016) but to improve the computation time of the data assimilation we coupled the model with the Parallel Data Assimilation Framework (PDAF; Nerger and Hiller, 2013). Perturbing the calibration parameters of the model as well as temperature and precipitation as forcing data, we generate 30 ensemble members and apply an Ensemble Kalman Filter, which is tuned with a forgetting factor of 0.8 (Evensen, 2003). During the assimilation of the original and

relocated dataset, we update all ten compartments of the WaterGAP, while we do not update the lakes and reservoirs when considering the removed dataset. To enable a fair comparison of the influence of the correction dataset, we use the same propagated error information for all three datasets for assimilation.



Here results are shown for the Alton subbasin only, which is the smallest of the major Mississippi subbasins (448,669 km$^2$) but located close to the Great Lakes, i.e. here the GRACE correction is particularly important to remove leakage-in effects
originating from large storage variations in the Great Lakes.

Running the WaterGAP model for all ensemble members without including observations is called Open Loop Simulation (OLS). Combining TWSA observations with the model in applying data assimilation (DA) pulls the model simulations towards the observations based on the uncertainties of simulations and observations (Fig. 10a). Figure 10b shows DA results using the different observations (original/removed/relocated) for the Alton subbasin. To analyze the effect of applying different
corrections to GRACE data on the assimilated results, we additionally compare linear trends derived over the time span under investigation for different runs against model-based results OLS (Tab. 1). In general, all the assimilation runs (DA) yield increased linear TWSA trends compared to the OLS emphasizing the influence of observations. In the Alton subbasin, the assimilation of the original GRACE observations introduces a spurious positive mass trend, which is assumed to not originate from storage increase within the subbasin itself, but to be caused by leakage due to a water storage increase in the Great Lakes,
particularly in the nearby Lake Superior and Lake Michigan. Subtracting the lake correction ("DA removed") from the GRACE data before assimilation prevents this trend from appearing in the model output. Re-adding the surface water storage before assimilation ("DA relocated") only slightly increases the trend again, which is another indication, that a large part of the trend introduced by the uncorrected GRACE data is caused by the influence of the Great Lakes. For the remaining subbasins, the influence of removing the lake/reservoir signal from GRACE has a smaller influence, details can be found in the Supplement
(Fig. S7). These findings emphasize the importance of applying such corrections to GRACE data not only for areas covered by water bodies with strong signals, but also for neighboring grid cells affected by strong leakage-in effects like the Alton basin.

| Subbasin | OLS | DA - GRACE | DA - Removed | DA - Relocated |
| --- | --- | --- | --- | --- |
| **Hermann** | 1.7 | 6.4 | 5.8 | 6.1 |
| **Alton** | 0.8 | 2.5 | 0.9 | 1.1 |
| **Metropolis** | -1.7 | -0.9 | -0.4 | -0.7 |
| **Vicksburg** | -17.5 | -8.3 | -8.0 | -8.3 |

**Table 1: Linear trends in mm EWH per year computed for 2003/01 - 2016/12 based on model-based results (OLS) and when assimilating (DA) original/removed/relocated GRACE-TWSA.**



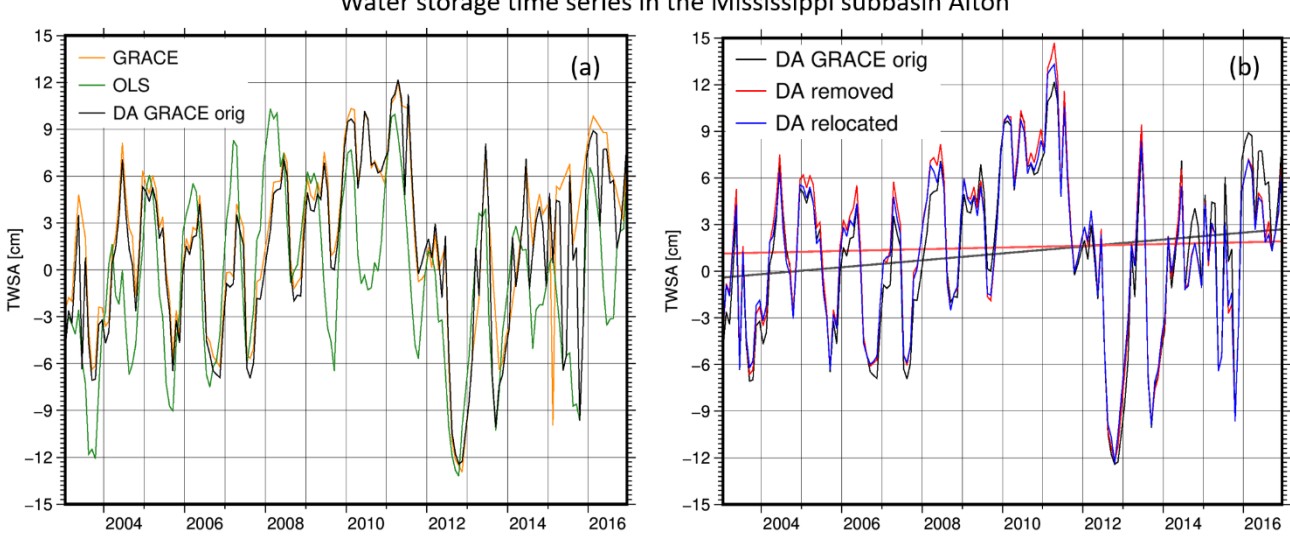


**Figure 10: Time series of TWSA in the Alton subbasin: original GRACE observations (orange), Ensemble mean of open loop WaterGAP simulations (green), model results after assimilating the original GRACE-derived TWSA (without applying any corrections, black), model results after assimilating the removed product (red) and after assimilating the relocated product (blue).**

### 4.3 Validation of RECOG-LR with GNSS

Global Navigation Satellite System (GNSS) time series contain signals resulting from surface mass redistribution: atmospheric pressure, non-tidal oceanic loading, and hydrological changes. Vertical displacement due to hydrological loading can be predicted by calculating the elastic response of an Earth model to the TWS changes. Previous studies showed good agreement between modelled deformation from GRACE-TWS and vertical displacement observed by GNSS (e.g., Springer, et al. (2019), Tregoning, et al. (2009), van Dam, et al. (2007)).

Here we use residuals from the ITRF2014 stacking (Altamimi et al., 2016) resulting from the second reprocessing campaign (repro2) by the International GNSS Service (IGS) (Rebischung et al., 2016) to validate the GRACE lake/reservoir correction data product RECOG-LR. The station velocities and discontinuities have been carefully removed from the GNSS time series. To be consistent with GRACE-TWS, the effects of atmospheric and non-tidal oceanic loading are subtracted from the time series using the AOD1B product (Flechtner et al., 2015). The displacements due to non-tidal oceanic and atmospheric loading

are calculated at daily epochs to be consistent with the GNSS observations. GRACE-TWS anomalies are converted into displacements in the spatial domain using spatial convolution of point mass load Green's function (Farrell, 1972) in the center-of-figure (CF) frame using a set of high-degree load Love numbers determined by Wang, et al. (2012) for the Preliminary Reference Earth Model (PREM) (Dziewonski & Anderson, 1981). Finally, the daily GNSS displacements are averaged to monthly intervals before removing the temporal mean and linear trend from both GNSS and GRACE-derived (modelled)

displacements.





Here the GNSS time series are used to assess the impact of the lake/reservoir correction on GRACE data. To get an idea of the magnitude of its influence, we first show the temporal RMS of the forward-modelled RECOG-LR signal (Fig. 11a) for ITRF2014 GNSS sites around the Great Lakes. The RMS of the vertical displacements amounts up to 1.4 mm for station ALGO (~190 km away from the lake shore) and can reach higher values for other stations provided by NGL (Blewitt et al.,

2018) (not shown), such as 2.3 mm at station BAKU close to the Caspian Sea and 2.5 mm at CHB5 directly at the shore of the Lake Huron.

To investigate the agreement of observed and GRACE-derived displacement time series, we compute the reduction in temporal RMS after subtracting the modeled displacements from the observed time series:

$$RMS_{Reduction}(\%) = \frac{RMS_{GNSS} - RMS_{GNSS-GRACE}}{RMS_{GNSS}} \times 100$$

By removing the modelled displacement based on GRACE adjusted with RECOG-LR from the GNSS time series, all stations have their RMS reduced (Fig. 11b). This means for ITRF2014 GNSS sites around the Great Lakes we can explain some of the observed vertical displacement by TWS changes at most of the stations, with a stronger agreement at stations closer to the Great Lakes (ALGO and NRC1 stations).

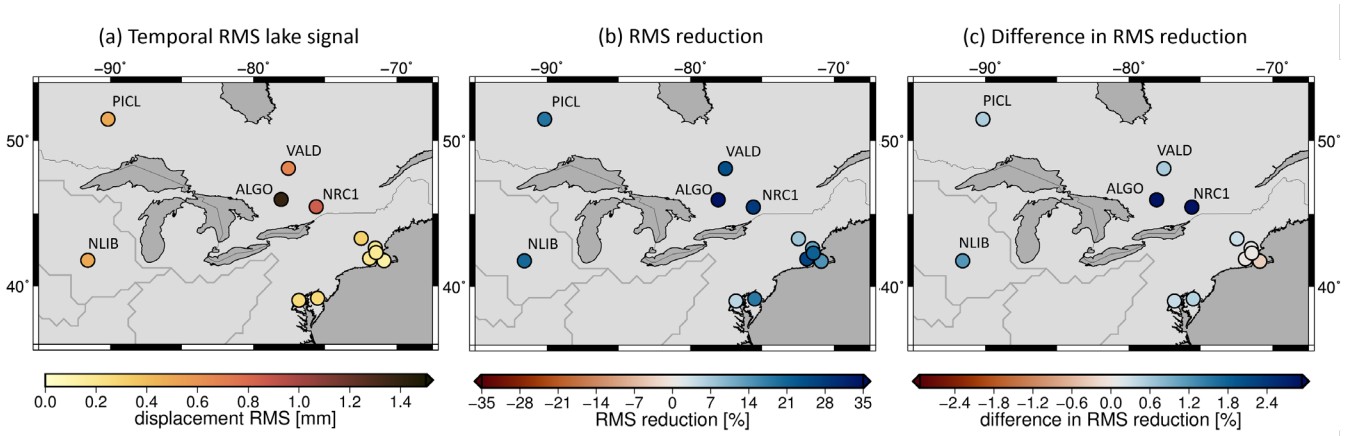

**Figure 11: Temporal RMS of the vertical displacement caused by the forward-modelled lake/reservoir signal of RECOG-LR (left), reduction of RMS of GNSS time series after subtracting modeled deformation (middle) and difference in RMS reduction when subtracting GRACE after removing and relocating RECOG-LR vs. using the uncorrected GRACE Signal.**

We show the performance of RECOG-LR by comparing the RMS reductions using (1) the original (unmodified) GRACE data

and (2) GRACE after subtracting and relocating the lake signal (Fig. 11c). All but one of the stations around the Great Lakes show a positive effect of the lake correction. The only station unimproved is located quite far from the lakes directly at the coast of the ocean and might primarily influenced by oceanic leakage as discussed in van Dam, et al. (2007). The largest improvements can be observed at stations ALGO (6.1%) and NRC1 (4.3%), both around 180-200 km away from Lake Huron. To put these numbers into perspective, a change in the Earth model used for the conversion of TWS to deformation, which has

previously been found to be relevant for GNSS analysis (Karegar et al. 2017), has a much smaller influence. The differences



in RMS reduction using, for example, different sets of load Love numbers amount to only <0.4%. Thus, from the above results, we are confident that correcting for the leakage effect of lake/reservoir water storage in GRACE time series can have a considerable positive effect on the comparison of GRACE and GNSS observations.

## 4.4 Hydrological drought detection with earthquake correction

Hydrological drought detection using GRACE data has been applied in various studies, for example in Houborg et al. (2012), Thomas et al. (2014), Zhao et al. (2017) and Gerdener et al. (2020a) for many regions on the Earth. If an earthquake removal is included, the results might lead to other drought detection results, which could have a significant impact on, for example, the decisions of policy makers. In this section, we show an example of the influence of the earthquake correction (Sec. 3.2) for detecting drought events in the Malaysian peninsular. To show the longer-term behaviour of droughts, the drought severity

index using accumulation (DSIA) used in Gerdener et al. (2020a) is computed. As typically done with meteorological indicators, the observable is accumulated for a chosen period $q$ ($\Delta TWS^{+}_{i,j,q}$) before its computation because we refer it to a duration of drought. The accumulation also serves as a running mean. The DSIA is then computed by

$$DSIA_{i,j,q} = \frac{\Delta TWS^{+}_{i,j,q} - \overline{\Delta TWS^{+}_{j,q}}}{\sigma^{+}_{j,q}}, \tag{7}$$

where $\Delta TWS_{i,j}$ is the accumulated TWS changes in year $i$ and month $j = 1, \dots, 12$, $\Delta TWS^{+}_{j,q}$ is the mean monthly accumulated TWS change, e.g. the mean over all Januaries, and $\sigma^{+}_{j,q}$ is the monthly standard deviation. Here, it is used to

identify hydrological drought events for an accumulation period of 6 months (DSIA6) over the Malaysian peninsular. Figure 12 shows the resulting DSIA6 time series using corrected (blue) and uncorrected (black) TWS changes. The uncorrected DSIA6 identifies a mainly moderate dry period in 2010 and 2011 and a severe period at the beginning of 2014. Both periods are also identified using the corrected DSIA6 but with different intensity. The period 2010/2011 is slightly more intense now and the drought in 2014 is extreme (e.g. Tan et al., 2017). Furthermore, the corrected DSIA6 shows exceptional drought in

2005, which was not identified with the uncorrected DSIA6. These findings are supported by, e.g., the EM-DAT database (EM-DAT, 2020) and Hashim et al. (2016), who also identified a drought over the Malaysian peninsular in 2005. However, in March 2005, a second earthquake (Nias earthquake) also occurred close to the Sumatra-Andaman region with a magnitude of M8.6. As stated by, for example, Broerse (2014), earthquakes with a lower magnitude are not always clearly visible in the data but it should be noticed that the Nias earthquake might still have possible influences on the time series analysis.



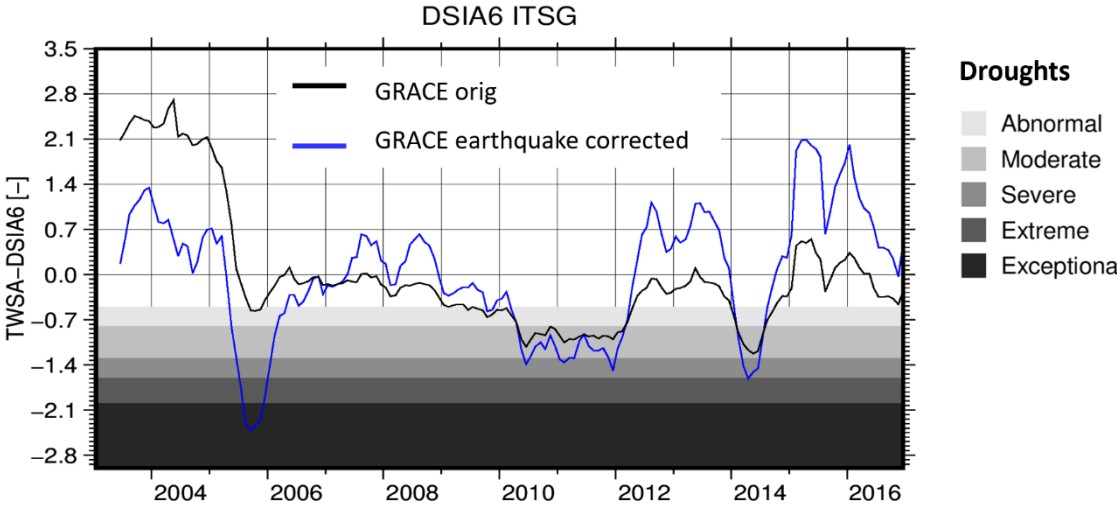

**Figure 12: GRACE-derived TWSA-DSIA6 over the Malaysian peninsular (West Malaysia) shown without (black) and with (blue) earthquake correction.**

## 4.5 Limitations

As mentioned in Sec. 2, static lake shapes were used to determine the lake volume and mass in further processing steps. Though this approximation works for most of the lakes with not too flat shores, there are some cases with either very flat shores and high water level variations or strong trends (or a combination of both phenomena), resulting in highly dynamic lake surface areas. This leads to larger errors when calculating with static lake shapes for the whole time period. A good example for this is the Aral Sea in Asia whose surface area shrunk to a fraction of its original size in the last couple of decades. It has thus not yet been included in the correction. The problem of missing consistent lake surface and/or volume variation has been addressed by several studies. Individual time series have been created e.g. by Singh et al. (2015) for Lake Mead and Aral Sea and Ni et al. (2017) and Ghansah et al. (2016) for Lake Volta. Regional (Zhang et al., 2014) or global investigations (Busker et al. (2019); Semmelroth (2019)) show that adequate results for lake water volume are best obtained by a linear regression model with an error between 3% and 15%.

Though RECOG-LR covers most of the major lakes around the world, some are not yet implemented in the dataset due to failed automatic matching between water level time series and lake surface area (e.g. Lake Athabasca, see also Sec. 2.1.3), insufficient water level time series due to ice coverage for major parts of the year (e.g. Lake Taymyr) or missing flyovers by altimeter satellites (e.g. Dead Sea, see also Sec. 2.1.1). More generally, we only consider surface water bodies that can be captured by satellite altimetry and this way underestimate the impact of surface water storage in regions with a large number of small lakes and dams, e.g. in the U.S. or in India. Additionally, changes in surface water volume also impact surrounding





groundwater storage (and thus GRACE data) by groundwater/surface water interactions (e.g. Bierkens & Wada 2019) which have not yet been considered in our data product.

## 5 Conclusions & Outlook

Leakage effects of surface water bodies and non-hydrology related mass change signals have a strong influence on water storage estimates from GRACE complicating its use for hydrological studies and specifically for calibration and data
assimilation. Volume change estimates from combining satellite altimetry with remote sensing information can be used to remove the effect of lakes and reservoirs from the GRACE data on a global scale, with particular benefit in regions close to big lakes/reservoirs or in regions with many smaller lakes and reservoirs. The earthquake signal (co-seismic and post-seismic), which masks hydrological variations in the vicinity of large earthquakes can be extracted directly from the GRACE time series. In this contribution, we introduced the first release of a new global correction dataset RECOG RL01 for removing both the
lake/reservoir storage (RECOG-LR) and the earthquake signal (RECOG-EQ) from the GRACE time series, while also offering the possibility to relocate the altimetry-derived mass change to its original surface water body outline.

Exemplary applications show that the correction product can reduce the signal variability (RMS) of the GRACE signal by up to 75% for the most prominent example of the Caspian Sea and that affected areas do not only include the lake areas themselves, but can extend for tens to hundreds of kilometers around the water bodies due to leakage. Special precaution has to be taken
when assimilating GRACE data into hydrological models in the proximity of large surface water bodies. In this context, the correction product is particularly valuable for models that do not include a surface water compartment at all, but the reduction of the leakage effect can also make it beneficial for models that do. For the example of the Alton subbasin of the Mississippi the leakage signal of the Great Lakes would cause an artificial mass increase in the assimilated model runs of WaterGAP, which can be prevented by subtracting and relocating the surface water storage before assimilation. A validation of the
corrected GRACE signal using observed vertical GNSS station displacements shows an improvement of the fit between GRACE and GNSS of up to 6% for stations at around 180 - 200 km distance from the Great Lakes. Applying the earthquake correction allows for the determination of several severe and one exceptional severe drought events in the Malaysian peninsular, which a GRACE-based drought indicator would otherwise have missed without first correcting for the earthquake signal. Therefore, depending on the application at hand, we recommend applying both RECOG-LR (globally) and RECOG-
EQ (especially when research is performed in areas that underlie large earthquakes) as a standard post-processing step for analyzing GRACE data.

Future improvements of the correction data product will introduce dynamic lakes shapes replacing the static lake surface areas used so far, which will enable the inclusion of surface water bodies with major area changes, such as the Aral Sea. The GRACE time series continuous thanks to GRACE-Follow-On and it is thus important to also provide a continuously updated surface
water body correction which will rely on continuously updated source data. An extension of the RECOG-LR to include further surface water bodies and a closure of existing data gaps as soon as new data is added to DAHITI will easily be possible thanks



to a fully automated process of matching lake IDs in the altimetry database with corresponding lake shapes and forward-modelling the water volume to filtered GRACE-like TWS. Thus RECOG-LR can be updated as soon as new data will become available.

So far, RECOG-LR focusses on lakes and reservoirs. However, there are other forms of surface water bodies whose effects on GRACE data are still disregarded and not yet covered by any correction. An example for this are rivers, especially river deltas of big river basins (e.g. Mississippi, Amazon, Congo) that are highly influenced by strong seasonal variations in water flux as well as influences by tides in the estuary. Such a correction would be especially interesting for hydrological modelling.

The correction data product can also be extended to cover additional geophysical phenomena. For example, since most
hydrological models do not include an explicit glacier compartment, it would be beneficial to extend the correction dataset to remove the glacier mass component from an independent glacier model or from remote sensing information before assimilating GRACE data into the model.

The presented correction products offer possibilities for more sophisticated data assimilation strategies. For example, GRACE data after removal of the lake/reservoir/earthquake correction can be assimilated into non-surface water compartments of a
hydrological model, while the altimetry-derived relocation dataset can be assimilated solely into the surface water compartment of models that explicitly incorporate this. The best way to use this information for data assimilation will have to be further investigated.

Further research for the earthquake correction should consider comparing different methods and should also analyze possible influences of earthquakes with a magnitude lower than 9.0 in more detail.


**Data availability.** RECOG includes (1) RECOG-LR with (1a) global gridded time series for the given time span with the correction values in total water storage for each grid cell and month, (1b) the same values in SH coefficients of degree and order 96 and (1c) the monthly gridded altimetry-derived water height for the relocation approach and (2) RECOG-EQ with the monthly gridded earthquake correction for TWS changes, containing the 2004 Sumatra-Andaman and the 2011 Tohoku
earthquake. The data has been uploaded to the PANGAEA database:

RECOG-LR: Deggim et al. (2020a), https://doi.org/10.1594/PANGAEA.921851; RECOG-EQ: Gerdener et al. (2020b, under revision), https://doi.pangaea.de/10.1594/PANGAEA.921923.

**Video supplement.** A time-lapse video of RECOG-LR showing global correction maps for each month has been uploaded to
the AV-Portal of the TIB Hannover: Deggim et al. (2020b), https://doi.org/10.5446/48188.

**Author contributions.** RECOG-LR was created by SD, based on ideas and suggestions from AE and LL and with help from LS. RECOG-EQ was created by HG. SM and IK (Sec. 2.1.1) and LE, CS, and DD (Sec. 2.1.2) provided input data. KS, OE and JK performed the data assimilation and ATS and TvD the validation with GPS. LS, AE and SD created the figures. All
authors contributed to and reviewed the manuscript.



**Competing interests.** The authors declare that they have no conflict of interest.

**Acknowledgements.** The support of the German Research Foundation (DFG) and the Luxembourg National Research Fund
(FNR) within the framework of the Research Unit GlobalCDA (FOR2630) is gratefully acknowledged.

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
