# Peer review of "RECOG RL01: Correcting GRACE total water storage estimates for global lakes/reservoirs and earthquakes"

_Earth System Science Data, 2020_

## Referee Comment (RC1) · Anonymous Referee #1 · 30 Oct 2020

General remarks: The manuscript introduces very interesting new corrections for the GRACE TWS data. While the manuscript is well written, for my liking it is too long. The problem is not, the often met, too less science and results, but the too much. Probably this would be enough for two publications... But this probably not changeable any more For a data publication is has to much validation/application but could spend more detail on the data set itself. Before Section 4 I thought the aim of the paper was about publishing two new data sets, after Section 4 I am not sure about the aim of the manuscript any more? Thus, please state more clearly what the aim is. Also the beginning of Section 4 could use a short introduction setting the frame for what is coming. About the sections in general: You should incorporate Section 3 in Section 2,

it recaps lots of information given in Section 2 and it would fit in there quite well. On the other hand, Section 4.5 should stand alone as it discusses the general limitations of the data sets and as it is now is a bit lost in Section 4. Finally, why do you not include the GRACE-FO time span already?

Overall, I think the manuscript is publishable after a major revision addressing my mentioned concerns.

Specific remarks: Abstract: First two paragraphs sound more like an introduction. Please shorten it. Section 2.1.1: Please provide more information on the Dahiti processing. Eg. did it uses the Ales+ retracker as suggested by the citation? What missions are used. About the accuracies of the time series, have they not improved since 2015. I would have expected this (or hoped for)! Section 2.1.2: Until Section 4.5 I thought you use time varying surface area data. Going back, I realized, this was what I expected in such a work. To my knowledge such data sets exist. Why do you not use them? Section 2.1.3: Why does a discretization on a 0.025° grid help to capture long and narrow reservoirs? Section 2.1.4: We found in recent investigations, that filtering a non GRACE data set with a DDK filter, introduces striping artefacts. This was most pronounced for a lake mass product over areas with sparse or no surface water data. Do you see something similar? Do you accept them to have a as similar as possible filtering of the data sets? I found paragraph l 200f and Figure 3 very confusing. The figure shows 3 different things (nicely in three different colors) but in the text only two are mentioned. Please make a stronger connection between text and figure (eg. colors in text and numbers in figure). Eq 5: The left-hand side of the equation is location depending, the right-hand side not? Section 4.3: I think I read this section now 10 times and I still do not know what you are showing in Fig 11. Is Fig. 11(a) the RMS between GPS time series and RECOG-LR? Then what is what in the equation? Eq 7: What is $q$? Also not clear that $\overline{TWS_{j,q}}$ is (and typo in the later?)

Minor remarks: l 47: replace Flechtner et al. 2016 to Kornfeld et al. 2019 (10.2514/1.A34326) l 85: You mean the WGHM? Then say so. Could you also give
a real citation not only a conference abstract? L 97 ff: Please also provide length of time series of different resolutions. L 198: surface water mass variations Eq 1: Up to which degree and order? L 246: When and where do you see 30cm correction? Fig 10: Even as a non-colorblind I cannot distinguish between black and blue line. Perhaps also put the two panels below each other to enlarge plots? L 399: I hope you have not used AOD1B Rl05, but Rl06. Then cite Dobslaw et al. 2017 https://doi.org/10.1093/gji/ggx302 l 436: You could also cite Boergens et al. 2020 https://doi.org/10.1029/2020GL087285

---

## Referee Comment (RC2) · Anonymous Referee #2 · 1 Jan 2021

Leakage problem is one of the key limitations of GRACE-like observations. Leakage between land and oceans has been examined in many previous studies, and this manuscript aims to correct other types of leakage associated with lakes and earthquakes. The data from this study would be useful for GRACE and hydrology communities. But before publication of this manuscript, I would like to raise some major issues.

Major points 1. Authors need to use much stronger validation of this leakage corrected GRACE data. They briefly compared crustal deformation between observation from GNSS and prediction from the leakage corrected data. Readers would like to see

actual time series between the two. If there are GNSS observations nearby other lakes than the Great Lakes, please show the results too. I also strongly recommend to examine horizontal displacements. Validation of the new data with independent observation should be very important.

2. Parametric fitting to earthquake signal would not be correct. Co-seismic signal should be okay because we know when step-like anomalies were. But post-seismic signal can be combined with other long-term variations particularly associated with TWS changes. Modeling of post-seismic deformation would be the best way for this like GIA correction. If the modeling is beyond the scope of this study, at least authors need to show that Cvpost are close to zeros for regions away from epicenters of earthquakes.

3. Isn't there steric effect in large lakes?

Minor Points

1. Sections 4.2 and 4.4 are about potential applications of the new data, and those sections are out of points for this manuscript. 2. Please show global maps for Figure7.

---

## Author Comment (AC1) · 14 Feb 2021

**Response to the reviewers' comments**
Deggim et al.

We would like to thank the two reviewers for their helpful comments that have helped us to improve our manuscript. Below we give our point-by-point answers (in blue) to the reviewers' suggestions (in black). The most important changes in the new version of the manuscript are

- a re-structuring of the manuscript sections to streamline the text following Reviewer 1
- an extension of the GNSS validation following Reviewer 2
- generally the clarification of several points that have been misleading in the original text.

Best regards,
Simon Deggim and Annette Eicker (on behalf of all co-authors)

**Response to Reviewer 1:**

**General remarks:**
The manuscript introduces very interesting new corrections for the GRACE TWS data. While the manuscript is well written, for my liking it is too long. The problem is not, the often met, too less science and results, but the too much. Probably this would be enough for two publications... But this probably not changeable any more. For a data publication is has too much validation/application but could spend more detail on the data set itself. Overall, I think the manuscript is publishable after a major revision addressing my mentioned concerns.

We would like to thank the reviewer for this opinion. Yes, the paper includes several different validation and application examples. However, we strongly believe that showing these different examples is necessary to demonstrate the value of the dataset and we therefore think that they are important for a proper discussion.

We also checked some papers recently published in ESSD and found several other examples that also contain rather extensive validation/application of the data sets. For example Kvas et al. 2021 (https://essd.copernicus.org/articles/13/99/2021/) have a detailed validation section of their gravity field model based on GNSS levelling and orbit residuals and Zhao et al. 2020 (https://essd.copernicus.org/articles/12/2555/2020/) show applications of their MODIS product. We therefore feel that ESSD does not have a strict policy against this and does not discourage validation and/or the showcasing of applications of the described data sets.

We would, therefore, prefer to keep the combination of data description and corresponding applications/validation together in one publication.

However, to make this intention clearer and to improve readability, the new version of the manuscript will

1) include a statement in the introduction and added a specific introductional paragraph at the beginning of the application/validation section.
2) be restructured according to the reviewer's suggestion (see answers below) to streamline the descriptions.

Before Section 4 I thought the aim of the paper was about publishing two new data sets, after Section 4 I am not sure about the aim of the manuscript any more? Thus, please state more clearly what the aim is. Also the beginning of Section 4 could use a short introduction setting the frame for what is coming.

Thank you very much for this comment. As stated above, the aim of the paper will be made clearer already in the introduction and we have added a short introduction on the intention behind the application/validation chapter at the beginning of the former Section 4 (now Section 3, see explanation on the restructuring of the manuscript below).

About the sections in general: You should incorporate Section 3 in Section 2, it recaps lots of information given in Section 2 and it would fit in there quite well. On the other hand, Section 4.5 should stand alone as it discusses the general limitations of the data sets and as it is now is a bit lost in Section 4.

Yes, thank you, we agree that it is a good idea to include the results of the correction data sets directly after the processing details to avoid repetition in the (former) sections 2 & 3. The revised version will be restructured accordingly and the description will be condensed to avoid duplications. Following the second suggestion, the discussion of the limitations of the data sets will be transferred to an individual section (now Section 4).

Finally, why do you not include the GRACE-FO time span already?

With the first release RECOG (RL01) we wanted to present a full data set for the (completed) GRACE mission. GRACE-FO data is currently being updated every month and thus a new correction data set would have to be created continuously. However, of course it is planned to include GRACE-FO data in upcoming releases of RECOG.

Abstract: First two paragraphs sound more like an introduction. Please shorten it.

Thank you for this opinion. However, we feel that a very short problem statement already in the abstract is necessary to set the scope for the following paragraph of the abstract, as not every reader of ESSD will be familiar with the specific challenges involved with GRACE data analysis (e.g. the leakage and signal separation issue). We would, therefore, prefer to keep the form of the abstract.

Section 2.1.1: Please provide more information on the Dahiti processing. Eg. did it uses the Ales+ retracker as suggested by the citation? What missions are used.

More information on the DAHITI processing are now included by giving information on the missions that are included (TOPEX, Jason-1/-2/-3, ERS-2, Envisat, SARAL, Sentinel-3A/-3B, ICESat, and Cryosat-2) and the applied retracker (Improved Threshold retracker). Additional details can be found in the cited literature on the DAHITI processing.

About the accuracies of the time series, have they not improved since 2015. I would have expected this (or hoped for)!

Yes, the accuracies have improved, but they are still in the same order of magnitude (few centimeters for larger lakes and RMSE of some decimeters for river crossings) and there has not been a more recent publication on this issue. Therefore, the given reference (Schwatke et al., 2015) is still valid.

Section 2.1.2: Until Section 4.5 I thought you use time varying surface area data. Going back, I realized, this was what I expected in such a work. To my knowledge such data sets exist. Why do you not use them?

Yes, we agree that using time varying surface areas would be ideal to further improve the accuracy of the correction data product. However, in this first version of RECOG we decided to use the more reliable static surface estimates, as they are less impacted by the uncertainties introduced by cloud coverage in the optical satellite images compared to monthly time series. We believe that this approximation works reasonably well for a large part of the surface water bodies included in our study and therefore already has a considerable benefit for correcting leakage effects in GRACE. Extreme cases, such as the Aral Sea, were explicitly excluded from the correction. We have added a short discussion on this in Section 4 ("Limitations"). For a future update of RECOG we plan on introducing the time-varying lake shapes exploiting a recently developed approach to fill the cloud-affected areas in the optical satellite images (Schwatke et al., 2019). This is mentioned in Section 5 ("Outlook").

Schwatke C., Scherer D., Dettmering D.: Automated Extraction of Consistent Time-Variable Water Surfaces of Lakes and Reservoirs Based on Landsat and Sentinel-2. Remote Sensing, 11(9), 1010, 10.3390/rs11091010, 2019

Section 2.1.3: Why does a discretization on a 0.025° grid help to capture long and narrow reservoirs?

The algorithm we use to detect whether a grid cell is within a given lake polygon only detects a water feature, if the center of the grid cell is within a water polygon. When directly using the 0.5 degree grid resolution, many water bodies would be missed completely. Thus we discretize the water bodies on the finer resolution and then upscale proportionally to 0.5 degree grid for the application to GRACE data. The updated version of the text will include more details on this.

Section 2.1.4: We found in recent investigations, that filtering a non GRACE data set with a DDK filter, introduces striping artefacts. This was most pronounced for a lake mass product over areas with sparse or no surface water data.Do you see something similar? Do you accept them to have an as similar as possible filtering of the data sets?

Thank you for pointing this out. Yes, it cannot be completely ruled out that very strong and localized non-GRACE-like signals can cause artificial striping after filtering. The effect of applying a DDK filter to non-GRACE signals (i.e. WGHM hydrological model output) have also already been illustrated in Kusche et al. (2009), Fig. 2-4, where it was shown that this depends on filter length. We assume that this might also play a role in causing some negative values in the RMS reduction map in the vicinity of the Caspian Sea in Fig.6b. We have added a comment on this in the text in the description of Fig. 6b (now Fig. 5b, see next comment). However, as you state, we decided to accept this to achieve an as similar as possible filtering of the data sets. However, alternatively filtered correction data sets, e.g. using an isotropic filter such a s the Gaussian filter) can easily be generated and are available upon request as stated in the text in Section 2.

Reference: Kusche, J., Schmidt, R., Petrovic, S. *et al.* Decorrelated GRACE time-variable gravity solutions by GFZ, and their validation using a hydrological model. *J Geod* **83,** 903–913 (2009). https://doi.org/10.1007/s00190-009-0308-3

I found paragraph I 200f and Figure 3 very confusing. The figure shows 3 different things (nicely in three different colors) but in the text only two are mentioned. Please make a stronger connection between text and figure (eg. colors in text and numbers in figure).

We have decided to remove the figure from the already quite long paper, as it does not seem to help for understanding. Instead we have made the corresponding text and equations more concise.

Eq 5: The left-hand side of the equation is location de-pending, the right-hand side not?
Thank you for pointing this out. We have corrected the equation and have included the location-dependency also on the right side of the equation.

In Section 4.3: I think I read this section now 10 times and I still do not know what you are showing in Fig 11. Is Fig. 11(a) the RMS between GPS time series and RECOG-LR? Then what is what in the equation?
We have updated the text to better explain what the values in the equations mean and what is shown in the figures.

Eq 7: What is q? Also not clear that overline(TWS_j,q) is (and typo in the later?)
As it was originally mentioned in the text, q is the period, over which the observable (here TWS) is accumulated. We have added an example for q = 3 months, which shows that the accumulated TWS changes for March 2003 would be the sum of TWS changes of January, February and March. We hope this explains it more clearly now. We have also corrected the typo in the text (thank you for pointing this out!): $\Delta TWS^+_{j,q}$ must is the mean monthly => $\overline{\Delta TWS^+_{j,q}}$

**Minor remarks:**
l 47: replace Flechtner et al.2016 to Kornfeld et al.2019(10.2514/1.A34326)
Thank you! We have added the reference to Kornfeld et al. (2019).

l 85: You mean the WGHM? Then say so. Could you also give a real citation not only a conference abstract?
Yes, of course, good point. We have included the direct reference to the model and have updated the citation to Müller-Schmied et al. (2014).

L 97 ff: Please also provide length of time series of different resolutions.
We have added the launch date of the different missions in the text to indicate the time spans for which the different resolutions are available.

L 198: surface water mass variations Eq 1:Up to which degree and order?
Directly above Eq. (1) it is mentioned that the maximum degree and order 96 was chosen for the expansion. To make this more clear we have additionally mentioned it for Eq. (2).

L 246: When and where do you see 30cm correction?
A correction of ~30cm can be seen for individual months in the vicinity of surface water bodies with very strong water storage change, such as the Great Lakes or the Caspian Sea. In the figure below (to be included in the Supplement and referred to in the text) we show two exemplary time series of RECOG-LR in such locations and the map of the correction in one specific month (2016-04). Fig. 4 of the paper only shows the seasonal amplitude (which is of course smaller), but the superposition with trend and interannual variations leads to larger corrections in individual moths.

[Figure]

*Fig. 1: Time series of RECOG-LR for two exemplary stations.*

Fig 10: Even as a non-colorblind I cannot distinguish between black and blue line. Perhaps also put the two panels below each other to enlarge plots?
We have changed the color of the blue line to make it better distinguishable from the black line.

L 399:I hope you have not used AOD1B Rl05, but Rl06. Then cite Dobslaw et al. 2017 https://doi.org/10.1093/gji/ggx302
Thank you for this correction. Yes, of course we have used AOD RL06 and we have now updated the citation.

l 436: You could also cite Boergens et al. 2020 https://doi.org/10.1029/2020GL087285
We have included the reference.

**Response to Reviewer 2:**

**General remarks:**
Leakage problem is one of the key limitations of GRACE-like observations. Leakage between land and oceans has been examined in many previous studies, and this manuscript aims to correct other types of leakage associated with lakes and earthquakes. The data from this study would be useful for GRACE and hydrology communities. But before publication of this manuscript, I would like to raise some major issues.

Authors need to use much stronger validation of this leakage corrected GRACE data.
We would like to thank the reviewer for this comment. We agree that a thorough validation of the correction data product is important. But we also acknowledge that this is challenging, as reliable independent observations of total terrestrial water storage (TWS) are largely missing. Independent measurements of hydrological quantities only refer to individual water storage compartments (e.g. in-situ soil moisture or groundwater measurements), there is no hydrological observation technique for TWS. Also terrestrial gravimetry (which in principle

does sense TWS variations) is challenging, because it is dominated by local mass effects. Furthermore, only very few monitoring sites of continuous gravity measurements exist and, to our knowledge, none of them is located close to large surface water bodies. GNSS station displacements are the only globally distributed observation type, which directly relates to TWS changes on spatial scales comparable to GRACE. We have, therefore, decided to extend the GNSS validation.

If there are GNSS observations nearby other lakes than the Great Lakes, please show the results too.

Following this suggestion, we have now additionally computed the influence of the lake correction for the available ITRF stations in the region of the Caspian Sea (NSSP and TEHN) and of Lake Victoria (NURK and RCMN) and will add the numbers to the paper. Again, for all these stations the RMS reduction (i.e. representing the fit between GRACE and GNSS) improves when applying the leakage correction, even though the stations are again located between 100-400 km away from the lake shores. The improvements might be small (0.2-2.7%), but systematically positive. Improvements of a few percent (up to 6% as shown for station AGLO) might seem small, but also many other corrections and processing choices that we apply in GNSS and other geodetic techniques lead to even smaller influences (e.g. the choice of different Love numbers (<0.4%) as discussed in the paper, or switching from one atmospheric loading correction to another (~2%, see Li et al, 2020). And we nevertheless spend the effort.

Furthermore, we would like to point out that it is not the intention of our study to investigate the influence of surface water storage on crustal movement directly due to lakes/reservoirs (e.g. directly at the shore), which has already been done by many studies, e.g. Gahalaut et al. (2017). Since the lake level in these studies represents the main signal, it is straightforward that the time series is dominated by it. However, we show in our study that even the ITRF sites, which are intentionally located far away from big reservoirs to avoid such a disturbance, are still influenced by the lake signal. And also here correcting leakage signals in GRACE and relocating the lake signal to its original location can improve the fit between GNSS and GRACE. Of course the lake influence is much smaller at several hundreds of kilometers away from the shore.

Li C, Huang S, Chen Q, Dam Tv, Fok HS, Zhao Q, Wu W, Wang X. Quantitative Evaluation of Environmental Loading Induced Displacement Products for Correcting GNSS Time Series in CMONOC. Remote Sensing. 2020; 12(4):594. https://doi.org/10.3390/rs12040594

Gahalaut, V. K., Yadav, R. K., Sreejith, K. M., Gahalaut, K., Bürgmann, R., Agrawal, R., ... & Bansal, A. (2017). InSAR and GPS measurements of crustal deformation due to seasonal loading of Tehri reservoir in Garhwal Himalaya, India. *Geophysical Journal International*, *209*(1), 425-433.

They briefly compared crustal deformation between observation from GNSS and prediction from the leakage corrected data. Readers would like to see actual time series between the two.

Following the reviewer's suggestion, we have additionally plotted the time series for three ITRF stations (ALGO and NRC1 in the Great Lakes region and RCMN around Lake Victoria), see figure below. From this plot, it can be observed that the RECOG-corrected time series (red) fit slightly better to the GNSS-observed displacements (black) than the original GRACE data (blue). This can be seen as evidence that with the corrected GRACE data sets, we are able to better explain the loading signal at the GNSS stations. However, of course the up to 6%

improvement in RMS reduction caused by the RECOG correction is not easy to detect from the time series plot. This is the reason why we prefer to show the aggregated information of the RMS reduction for several stations in the paper. But, following the reviewer's suggestions, we will include the time series plot in the Supplementary Material and refer to it in the text.

[Figure]

Vertical station displacements at ITRF2014 GNSS stations

Fig. 2: Time series of station displacements observed by GNSS (black) and modelled from the original GRACE time series (blue) and after applying the RECOG-LR correction (red).

I also strongly recommend to examine horizontal displacements. Validation of the new data with independent observation should be very important.

We would like to thank the reviewer for this opinion. However, very few studies have related horizontal station movement to GRACE and many point out the difficulties involved with this (Fu et al. 2013, Tregoning et al. 2009, and others). The reasons for this are mainly that horizontal displacements due to hydrological loadings are much smaller than vertical ones and have been found to be systematically underpredicted and out of phase (Chanard et al., 2018). Furthermore, horizontal station displacements are not independent from the vertical displacements, since one has to rely on a loading model as well. Based on these reasons, we believe that horizontal station observations would add little additional information and we thus refrain from this request. We will, however, included a short discussion on the reason for using only vertical displacements in the text.

Chanard, K., Fleitout, L., Calais, E., Rebischung, P., & Avouac, J. P. (2018). Toward a global horizontal and vertical elastic load deformation model derived from GRACE and GNSS station position time series. *Journal of Geophysical Research: Solid Earth*, *123*(4), 3225-3237.

Fu, Y., Argus, D. F., Freymueller, J. T., & Heflin, M. B. (2013). Horizontal motion in elastic response to seasonal loading of rain water in the Amazon Basin and monsoon water in Southeast Asia observed by GPS and inferred from GRACE. *Geophysical Research Letters*, *40*(23), 6048-6053.

Tregoning, P., Watson, C., Ramillien, G., McQueen, H., & Zhang, J. (2009). Detecting hydrologic deformation using GRACE and GPS. *Geophysical Research Letters*, *36*(15).

Parametric fitting to earthquake signal would not be correct. Co-seismic signal should be okay because we know when step-like anomalies were. But post-seismic signal can be combined with other long-term variations particularly associated with TWS changes. Modeling of post-seismic deformation would be the best way for this like GIA correction. If the modeling is beyond the scope of this study, at least authors need to show that Cvpost are close to zeros for regions away from epicenters of earthquakes.

Thanks for this comment. We believe that both methods are possible and widely used and demonstrated in the community: The extraction of the earthquake signal by using (1) a parametric fitting and (2) earthquake models. The two methods have different advantages and disadvantages that one should be aware of.

Big advantage of (1) is that we use real observation data. It is right that there is no method that perfectly extracts the post-seismic component from the data and the method might partially identify different long-term signals as post seismic relaxation. However, this part should not be very large because of the step function, which is not only introduced for the co- but also for the postseismic part (see $H_{t_v}$ in Eq. 5). Fig. S6 in the Supplementary Material shows this very nicely on the temporal domain, before the beginning of the earthquake the correction (co- and post seismic part together) shows values equal or close to zero. For the spatial domain we described that we limit the effect of earthquake correction locally with the Gauß filter to the earthquake regions so it is per definition set that there will be only values close to zero for regions far away from the epicenters. This weighting is applied to the complete correction field and not only Cvpost. Figure S5 visualizes this effect regionally as well. Using models (2) would have the disadvantage that they do not perfectly represent the reality because they are based on underlying assumptions and influenced by uncertainties of input data. Models rely on dislocation parameters, fault geometry and background rheological models and parameters which are all not known in reality. And these parameters are typically made to fit observed seismic moment tensor and (possibly) horizontal displacements, typically not in the epicenter region. It is known that they often then do not fit GRACE at all (see Einarsson et al. 2010, which is cited in the paper).

Both methods can work but we believe that for the intention of the paper, which is presenting correction data sets for the real GRACE data, we should rather use the data itself for correction before introducing new errors derived from models. We have added a comment on this to the text.

Isn't there steric effect in large lakes?

This is a good point, which we agree should be mentioned in the paper. Yes, there are steric effects occurring in large lakes. However, due to a stronger stratification of lakes (compared to oceans) because of less vertical mixing a much smaller variability of the steric term can be expected over time. We expect relevant effects only in the largest surface water bodies. Studies have been carried out for the Caspian Sea, where Chen et al. (2017) find the steric effect of the seasonal amplitude to be in the range of 1/3 of the altimetric signal with an only

negligible influence on the trend. However, due to a lack of temperature and salinity data, they transfer oceanic temperature/salinity from a location in the same latitude and state that this can only produce a very rough estimate, since different thermal heating processes between the Caspian Sea and Northern Atlantic must be assumed. A similar order of magnitude of the steric influence is found in Loomis & Luthcke (2017). We consider the Caspain Sea as an extreme case, much smaller effects are expected in smaller water bodies. However, detailed studies on this are missing and no global data exist. Therefore, it is not feasible to model the steric effect in a global data product such as RECOG-LR. However,the revised version of the manuscript will include a discussion on this issue in the text and this limitation of the product will be pointed out.

Chen, J. L., Wilson, C. R., Tapley, B. D., Save, H., & Cretaux, J. F. (2017). Long-term and seasonal Caspian Sea level change from satellite gravity and altimeter measurements. Journal of Geophysical Research: solid earth, 122(3), 2274-2290.

Loomis, B. D., & Luthcke, S. B. (2017). Mass evolution of Mediterranean, Black, Red, and Caspian Seas from GRACE and altimetry: accuracy assessment and solution calibration. Journal of Geodesy, 91(2), 195-206.

**Minor remarks:**
Sections 4.2 and 4.4 are about potential applications of the new data, and those sections are out of points for this manuscript.
We would like to thank the reviewer for this opinion. Yes, the paper does include several different validation and application examples. However, we strongly believe that showing these different examples is necessary to demonstrate the value of the data set and that they are mandatory for a proper discussion. We would, therefore, perfer to keep the combination of data set description and corresponding applications/validation together in one publication.

Please show global maps for Figure 7.
We have additionally created the global maps. However, as you can see below, the global trend maps are very much dominated by, e.g., mass loss trends in the polar ice sheets and by groundwater changes in several regions. Therefore, with the exception of the Caspian Sea the trends in surface water storage are only marginally visible in the global maps, which is why we feel the information content of these global maps is rather limited. However, since it might nevertheless be of interest to some readers, we will include this figure in the Supplement.

[Figure]

*Fig. 3: Global map of trends in the original GRACE data (top left), after removing the surface water correction RECOG-LR (top right) and after relocating the surface water storage (bottom).*

---

## Editor Comment (EC1) · Christian Voigt (Editor) · 15 Feb 2021

Dear authors, Thank you for your reply to the comments of the referees. Please upload a revised version of your manuscript to the ESSD system along with your detailed point-to-point reply.

Best regards, Christian Voigt (Topical Editor)

---

## Author Response (AR2)

**Response to the reviewers' comments (rev 2)**

Deggim et al.

We would like to thank the editor Christian Voigt and the two reviewers for their feedback and comments. Following their suggestions, we have made the following changes:

- L. 246-247: We added further information about the radius of spatial radial Gaussian window.
- L. 291: We added the missing reference for Glacial Isostatic Adjustment (A et al., 2013).
- Some minor rephrasing and spelling corrections were carried out, see marked-up manuscript version.

Best regards,
Simon Deggim and Annette Eicker (on behalf of all co-authors)

---

## Author Response (AR3)

**Response to the reviewers' comments (rev 3)**

Deggim et al.

We would like to thank the editor Christina Voigt for his suggestions for technical corrections. According to the points stated in the minor revision comment, we have changed the manuscript as follows:

- Reordering of references that were not in alphabetical order.
- DOIs were added to all references with available DOIs. For Döll et al. 1999, no DOI seems to be available.
- Minor corrections were made to the references to align with the desired reference style.

Best regards,
Simon Deggim and Annette Eicker (on behalf of all co-authors)